# Knowledge and remaining gaps on the role of animal and human movements in the poultry production and trade networks in the global spread of avian influenza viruses – A scoping review

Claire Hautefeuille[1,2,3]*, Gwenaëlle Dauphin[3], Marisa Peyre[1,2]

1 CIRAD, UMR ASTRE, Montpellier, France, 2 ASTRE, CIRAD, INRA, University of Montpellier, Montpellier, France, 3 CEVA Santé Animale, Libourne, France

* claire.hautefeuille@cirad.fr

**Data Availability Statement:** All relevant data are within the manuscript and its Supporting Information files.

## Abstract

Poultry production has significantly increased worldwide, along with the number of avian influenza (AI) outbreaks and the potential threat for human pandemic emergence. The role of wild bird movements in this global spread has been extensively studied while the role of animal, human and fomite movement within commercial poultry production and trade networks remains poorly understood. The aim of this work is to better understand these roles in relation to the different routes of AI spread. A scoping literature review was conducted according to the PRISMA guidelines (Preferred Reporting Items for Systematic Reviews and Meta-Analyses) using a search algorithm combining twelve domains linked to AI spread and animal/human movements within poultry production and trade networks. Only 28 out of 3,978 articles retrieved dealt especially with the role of animal, human and fomite movements in AI spread within the international trade network (4 articles), the national trade network (8 articles) and the production network (16 articles). While the role of animal movements in AI spread within national trade networks has been largely identified, human and fomite movements have been considered more at risk for AI spread within national production networks. However, the role of these movements has never been demonstrated with field data, and production networks have only been partially studied and never at international level. The complexity of poultry production networks and the limited access to production and trade data are important barriers to this knowledge. There is a need to study the role of animal and human movements within poultry production and trade networks in the global spread of AI in partnership with both public and private actors to fill this gap.

**Funding:** This study was funded by Ceva Santé Animale (https://www.ceva.com/en/). The funders had no role in study design, data collection and analysis, decision to publish, or preparation of the manuscript.

**Competing interests:** This study was funded by Ceva Santé Animale (https://www.ceva.com/en/). Claire Hautefeuille and Gwenaëlle Dauphin are employees of Ceva Santé Animale. CIRAD has a collaborative research agreement with Ceva as part of the private-public funded PhD scholarship of Claire Hautefeuille. This does not alter our adherence to PLOS ONE policies on sharing data and materials. The funders had no role in the study design, data collection and analysis, decision to publish, or preparation of the manuscript.

# Introduction

World poultry meat production increased by 21.3 million tonnes between 2010–2017 [1]. The United States, Brazil, China and the European Union are the biggest poultry producers in the world [1] with chicken accounting for the most produced meat worldwide since 2016 [2]. Pork and poultry are the most consumed meats worldwide, with about 16kg per capita [2]. Poultry also represents the biggest meat trade [2]. The main poultry meat exporting countries are Brazil, United States, European Union and Thailand. Whereas China, Japan, Mexico and Saudi Arabia import the highest volume of poultry meat [1].

Low and high pathogenic avian influenza viruses (LPAI and HPAI respectively) are disseminated worldwide. While AI viruses were identified at the end of the nineteen century the number of outbreaks caused by HPAI has shown an upward trend since the last years of the twentieth-century [3]. In 1996, A/goose/Guangdong/1/1996 (H5N1), the precursor of currently circulating H5N1 HPAI viruses was identified in farmed geese in southern China. Since then the large majority of HPAI outbreaks around the world have been related to these H5N1 HPAI viruses [4]. AI outbreaks have mainly been reported in Asia, and to a lesser extent in Africa, North America and Europe [5]. No outbreaks were reported on the South American continent between 2010 and 2016. H5N1 remains the most dominant AI virus subtype, among reported outbreaks; however, it is worth mentioning that only outbreaks caused by HPAI and LPAI H5-H7 have to be officially notified to the World Organisation for Animal Health (OIE). During the period 2010–2016, the reported outbreaks mainly referred to commercial farms, followed by wild bird species and backyard domestic poultry [5]. AI introduction and global dissemination via wild birds have been extensively studied [6–11]. Similarly, the subsequent dissemination and spread of AI within or between farms is highly documented [12–17].

Only a limited number of studies have looked into the respective roles of the different poultry production networks in the emergence and spread of AI. A recent review has highlighted that intensive poultry production networks increase the probability of mutation of LPAI to a HPAI and shown that the type of mutation (conversion or reassortment) is influenced by the type of production networks (developed or transitioning) [18]. Two studies in Egypt and China have shown links between the increase of the poultry production and the increase of AI outbreaks [19,20]. One of these studies made a further analysis suggesting that this link could be applied at the global level [20]. More generally, relationships between economic growth, globalization, emerging and global spread of diseases including AI have been described [21,22].

Limited attention has been drawn to the risk of local and international dissemination of AI via poultry production and trade networks. Poultry production networks encompass all actors of commercial poultry production from hatchery to slaughterhouse, including commercial farms, and the links between them. They involve live bird movements at different stages of production (e.g. hatching eggs, day-old chicks, adult birds, etc.) along with human worker movements. These movements can take place at local, national but also international level. Poultry trade networks encompass all actors of the commercial live poultry trade between poultry production networks or from poultry production networks to consumers. They involve live bird movements along with human trader movements. As for the movements of poultry production networks, these can take place at local, national and international level. The potential spread of HPAI (via live birds or fomites) within these poultry production and trade networks needs to be taken into consideration at all levels. The first objective of this work was to list the identified routes of AI spread within poultry production and trade networks. The second objective was to improve the understanding of the current knowledge and

gaps on the role of these animal, fomite and human movements within the poultry production and trade networks on the global AI spread, to inform the need for further research.

## Materials and methods

### Protocol

A scoping literature review was conducted according to the PRISMA-ScR guidelines (Preferred Reporting Items for Systematic Reviews and Meta-Analyses extension for Scoping Reviews) [23,24] (S1 Table). This study followed the methodology proposed by Arksey and O'Malley [25]: identifying the research question, identifying relevant references, selecting references, charting the data, collating, summarizing and reporting the results.

### Identifying research questions

The scoping review was conducted to answer the following research questions:

1. What are the identified routes of AI spread?

2. What is the role of animal, human and fomite movements in the global spread of AI within poultry production and trade networks?

Studies looking at the spread of all AI subtypes (including both LPAI and HPAI) were included. Global spread through wild birds and more specifically migratory birds was not included in this review as it has been the focus of a recent review [10]. Moreover, our review focused only on animal health and not on human health.

### Identifying relevant references

**Eligibility criteria.**   This literature search included references published between January 1975 and May 2019 (inclusive), in the English language and with available abstracts.

**Information sources.**   This study used five information sources (CAB Abstract, Web of Science, Medline, Scopus and Science direct databases) to identify references.

**Search.**   Twelve domains were included in the search, with several keys words for each, i.e. diffusion ("diffusion OR transmission OR spread"), emergence ("emergence OR introduction OR outbreak"), epidemiology ("epidemiology"), risk ("risk"), model ("model"), influenza ("influenza"), avian ("avian OR poultry OR duck OR chicken OR chicks OR geese OR turkey OR quail OR partridge"), network ("network OR organization OR value-chain"), production ("production OR company OR farm OR industry OR sector"), trade ("commercial OR trade"), movement ("traffic OR mobility OR movement") and domestic ("domestic"). The search algorithm was the following combination of these twelve domains: [(diffusion) OR (emergence) OR (epidemiology) OR (risk) OR (model)] AND [influenza] AND [avian] AND [(network) OR (production)] AND [(trade) OR (movement) OR (domestic)]. As an example, the search request used for Scopus on the 31 Mai 2019 was: (TITLE-ABS-KEY (diffusion OR transmission OR spread OR emergence OR introduction OR outbreak OR epidemiology OR risk OR model) AND TITLE-ABS-KEY (influenza) AND TITLE-ABS-KEY (avian OR poultry OR duck OR chicken OR chicks OR geese OR turkey OR quail OR partridge) AND TITLE-ABS--KEY (network* OR organization* OR value-chain OR compan* OR production* OR farm* OR industr* OR sector) AND TITLE-ABS-KEY (commercial OR trade OR traffic OR mobility OR movement OR domestic)).(S2 Table).

An additional search was performed using Google Scholar to identify any relevant references not published in peer-reviewed journals. This search was made following the recommendations of Haddaway et al. on literature search using Google Scholar [26]. The literature

search using Google Scholar is technically limited because of a specific and simple construction of the search algorithm with the possibility of using only one "AND" and a limited number of key words. Because of these limitations, a different search algorithm was used: ["avian influenza" AND [(network) OR (trade) OR (movement)]] with the use of the same key word domains as previously described. Moreover, as the references have to be imported manually, the removal of duplicated searches between literature databases and Google search was performed during the screening step to ease the process. Grey literature retrieved from personal contacts (e.g. FAO reports) and references identified through citations were also included in this analysis. All references retrieved from the scientific databases were imported into Zotero® version 5.0 and duplicate references were removed.

## References selection

The references were selected through a first screening phase, based on title, abstract and full text if necessary using the following exclusion criterion: 1) "references not on AI spread" to address the first research question. The remaining references were then selected through another screening phase on abstract and full text if necessary using the following exclusion criteria: 2) "references not within poultry production and trade networks" and 3) "references on risk factors without considering animal, human or fomite movements". A flow chart diagram of the inclusion selection process for publication in this study was developed based on the PRISMA approach (Fig 1).

## Data charting process and data items

A database template was developed in Microsoft Excel® version 2007 to extract the following data from the retrieved references from the first screening step: type of routes studied and type of risk factors studied. Another database template was developed in Microsoft Excel® version 2007 to extract the following data from the retrieved references from the second screening step: year, location, study objectives, study type, data source, method to analyse data, type of network studied, type of movement studied and results (results on AI spread, animal movements, human movements, on the role of these movements in AI spread, other results). A

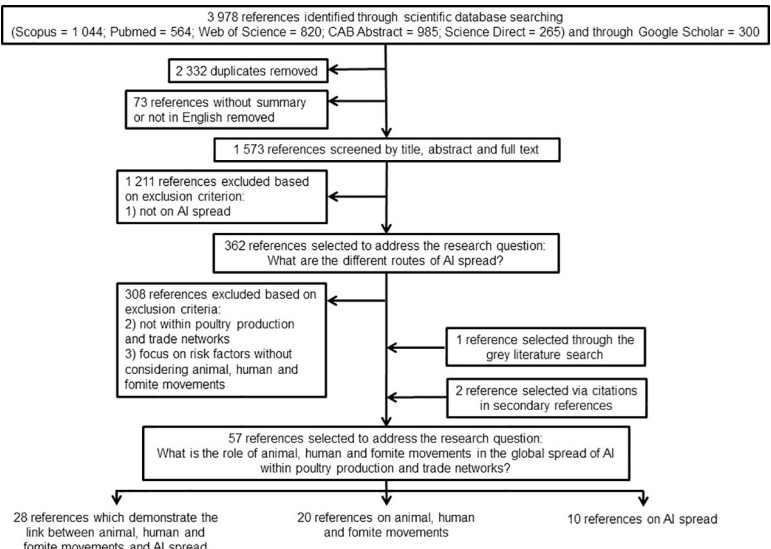

**Fig 1. Flow chart diagram of the study selection process for inclusion in this scoping review.**

descriptive analysis of the references linked to the review topic was performed at each screening step and described in this paper.

## Synthesis of results

References selected after the first screening step were used to describe the routes of AI spread between different compartments (commercial farm, production and trade network, wild birds, etc.). Among these selected references, those focused on risk were analysed separately as they provided information on the risk of the different AI pathways listed. These references focused on risk factors were classified according to the following compartments: international trade, national trade, poultry production network, commercial farm, backyard, environment and wild birds. The risk factors presented in these references were then categorised according to the risk of AI transmission analysed: risk of transmission within a compartment (e.g. within farm) or risk of transmission between compartments (e.g. from wild birds to farm). The result of this classification was used to build a figure on the different compartments involved in AI circulation and the transmission pathways between them. The remaining references before the application of the third selection criteria were classified between the different compartments considered on this figure. References selected after the second screening step were classified into three groups following the study objectives. Major results linked to the second research question were summarized.

## Results

This study retrieved 3,978 references from the scientific databases including 2,332 duplicates and 73 references removed because they were not written in English or were missing an abstract (Fig 1). Those 362 references were selected to identify the different AI spread pathways (first research question) (348 on HPAI spread, 12 on LPAI spread and 2 on both). References not related to poultry production and trade networks or on risk factors without considering animal, human and fomite movements (308) were removed (Fig 1). In addition to the remaining 54 references, three references from the grey literature and identified through citations were included at this stage. These 57 references (54 on HPAI spread, 1 on LPAI spread and 2 on both) were used to improve the understanding of the current knowledge and gaps on the role of these animal and human movements, including fomites, within the poultry production and trade networks in the spread of AI (second research question).

### Identified routes of avian influenza spread

Among the 362 selected references, 276 described the different AI spread pathways while 86, which focused on risk factors, addressed the risk of these different pathways. Most of the retrieved references related to AI spread pathways were looking at the poultry production and trade network level (49/276) and at the commercial farm level (84/276), as our literature search focused on poultry production and trade networks (Fig 2). Nevertheless, many of the articles identified were related to the spread of AI in wild birds (108 references, including 32 that discussed interactions between wild birds and commercial farms while the other references discussed AI spread within wild bird populations) and backyard (44/276). Only a limited number of references studied the role of international trade (5/276) and these references were also selected in this process [27–31].

A large number of references (86) focused only on risk factors including the risk between and intra-commercial farms and trade and production networks (Table 1). A limited number of references studied the risk at the interface between trade and production networks and backyard, environment and wild birds.

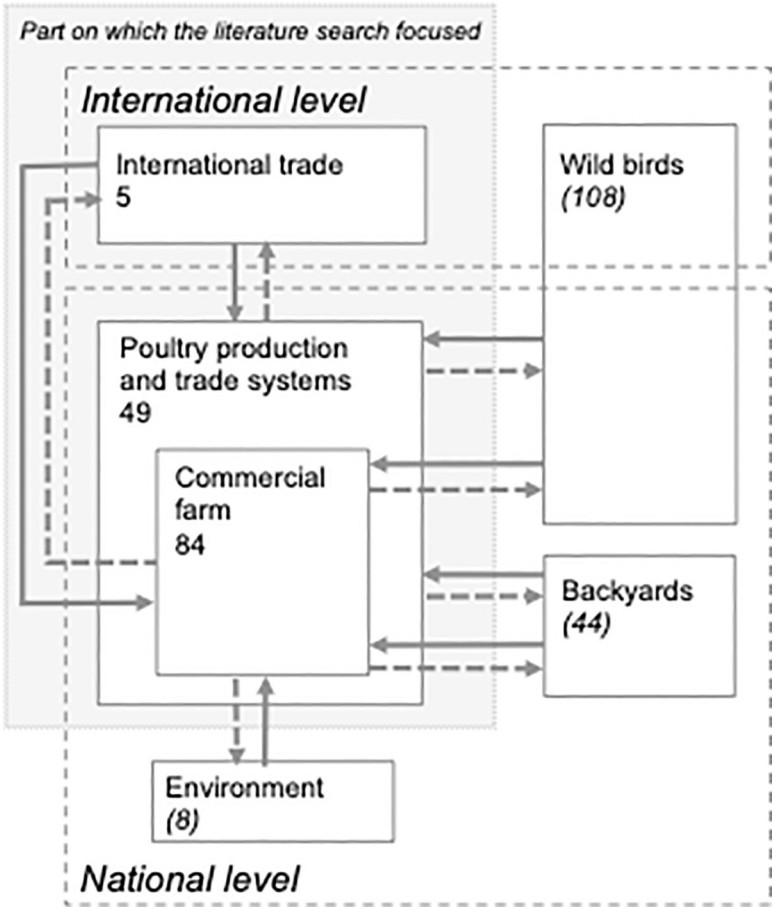

**Fig 2. The different compartments of avian influenza virus circulation and links between them: Demonstrated circulation (plain arrow); suspected circulation (dash arrow).** Number without bracket: Number of references identified on avian influenza (AI) spread at the compartment level. Number with bracket: Non exhaustive number of references identified on AI spread at the compartment level. Plain arrow: AI pathways from one compartment to another studied by at least one record on risk factors of AI spread selected by the literature search. Dash arrow: Possible AI pathways from one compartment to another: not studied by any record on risk factors of AI spread selected by the literature search.

**Table 1. Classification of the 86 references focusing on risk factors of avian influenza spread according to the compartment studied and/or the transmission pathway studied.** Light grey: less than 5 references, medium grey: between 5 and 20 references, dark grey: more than 20 references.

| To / From | International trade | National trade network | National production network | Commercial farm | Backyard | Environment | Wild birds |
|---|---|---|---|---|---|---|---|
| International trade | 0 | 1 | 4 | 2 | 0 | 0 | 0 |
| National trade network | 0 | 4 | 0 | 10 | 3 | 0 | 0 |
| National production network | 0 | 0 | 10 | 13 | 1 | 0 | 0 |
| Commercial farm | 0 | 0 | 0 | 28 | 0 | 0 | 0 |
| Backyard | 0 | 2 | 1 | 7 | NA | NA | NA |
| Environment | 0 | 0 | 0 | 13 | NA | NA | NA |
| Wild birds | 0 | 0 | 8 | 27 | NA | NA | NA |

NA: not applicable, when both compartments (origin and destination) were not included in the search question, i.e. backyard, environment, wild birds. 0: no reference on the 86 selected references on risk factors studied these transmission pathways.

## Animal and human movements and avian influenza viruses spread within poultry production and trade networks

Fifty-seven references remained after the application of the exclusion criteria 2) "references not related to poultry production and trade networks" and 3)"references on risk factors without considering animal, human and fomite movements" and the addition of references identified through citations as well as from the grey literature (Fig 1, Table 2, S3 Table).

These references can be divided into three groups according to their main objectives: studies which aimed to 1) demonstrate the link between animal, human and fomite movements and AI spread within poultry production and trade networks (28 references); 2) describe animal, human and fomite movements in a context of AI spread within poultry production and trade networks (but with no direct demonstration of the link) (20 references); 3) describe the AI spread within the production and trade networks (without making explicit links to animal,

**Table 2. Classification of the selected studies on animal and human movements and AI spread within poultry production and trade networks according to the type of network studied, the objectives, the type and location of the study.**

| Classification | Type of network studied | Objectives | Type of study | Location | References |
|---|---|---|---|---|---|
| Studies which aimed to demonstrate the link between animal, human, fomite movements and AI spread within the poultry production and trade networks | International trade network | To determine if the international spread of HPAI H5N1 was influenced by the poultry trade from infected countries | Modelling | Asia | [30] |
| | | To evaluate the risk of introduction and dissemination of HPAI through legal and illegal trade | Risk assessment | Ethiopia, Spain, Vietnam | [27,29,31] |
| | National or local trade network | To investigate association To improve knowledge on live bird trade To assess surveillance and control strategies To assess biosecurity practices | Modelling | China, Indonesia, Vietnam | [33–35] |
| | | | Network analysis | China, Pacific Islands, Vietnam | [33,36–40] |
| | National or local production network | To identify the role of commercial farms in the persistence and the spread of AI To assess surveillance and control strategies To assess the risk of AI spread | Modelling | France, Netherlands, UK, USA | [41–51] |
| | | | Network analysis | Korea, South Africa, UK | [51–53] |
| | | | Risk assessment | Australia, USA | [54–56] |
| Studies which aimed to describe animal, human and fomite movements in a context of AI spread within poultry production and trade networks (but with no direct demonstration of the link) | International trade network | To describe genetics value-chain | Descriptive analysis | Global | [28] |
| | National trade network | To describe poultry trade network To assess surveillance and control strategies | Network analysis | Bangladesh, Cambodia, Kenya, Mali, Vietnam | [32,57–61] |
| | | | Descriptive analysis | China, Vietnam | [62,63] |
| | National production network | To describe the poultry production network To assess surveillance and control strategies To assess the impact of a potential HPAI introduction | Network analysis | China, Egypt, Indonesia, Kenya, Nepal, Nigeria, | [64–70] |
| | | | Descriptive analysis | Australia, Switzerland, UK | [71–74] |
| Studies which aimed to describe AI spread within the production and trade networks (without making explicit links to animal, human and fomite movements) | National trade network | To analyse AI spread To assess surveillance and control strategies | Modelling | India, Vietnam | [32,75] |
| | National production network | | | Italy, France, Ghana, Netherland, Nigeria, USA, Vietnam | [76–83] |

UK, United Kingdom; USA, United States of America

human and fomite movements) (10 references) (Table 2). One of these studies described a live bird market network and studied AI spread within live bird markets but did not demonstrate the link between the two [32].

References on AI spread mostly used epidemiological modelling, while references on animal, human and fomite movements mostly used network analysis. Both approaches show the link between movements and disease spread: network analysis represents the connections between different units and assesses their relative importance in the network (centrality or connectivity) while disease modelling integrates such movements as parameters. Two studies actually combined both approaches [33,51]. This combination made it possible to use the connections described by network analysis as input parameters for the epidemiological model.

It is noteworthy to mention that network analysis studies mainly relied on data taken from field studies (e.g. cross-sectional interviews) and/or official data (e.g. from the Ministry of Agriculture or from national veterinary services) while epidemiological modelling mostly relied on official data (usually entered in national or global databases) and/or data from scientific literature. Only three studies included data obtained directly from the private poultry industry [42,48,54].

## Type of animal and human movements

From the 57 references, 48 studied animal and human movements (Table 2). Movements were described according the type of network studied (Table 2): international trade network, national trade network and national production network.

Only a limited number of studies looked at the international trade network and these only studied animal movements (Table 3): animal genetics trade at the global level [28], live chicken trade at the regional level (Asia) [30] and risk of disease introduction in a country through day-old chicks (DOC) [29] or adult bird imports [31]. One reference studied illegal international trade of DOC and adult birds between two countries linked to AI spread [27]. Another reference focused on the national trade network in Cambodia and briefly mentioned illegal trade from neighbouring countries [61].

At the national level, three types of movement were studied: animal, human and fomite movements. Overall, all types of production were studied, from breeders to production birds, including DOC and hatching eggs and a majority of domestic species (chicken broilers, layers, ducks, turkeys, ostriches, quails, indigenous birds). Animal movements are either movements linked to the trade network or to the production network. Three references looked at movements related to the specific structure of a production network: the fattening duck production in France [44], free-grazing duck production in Vietnam [58] and ostrich production in South-Africa [53]. These production networks require a change of farm or location for the different stages of growth of the birds, which involves a lot of between-farm movements. One reference looked at movements linked to poultry shows [72]. In regard to fomites linked to AI spread, the majority of studies focused on feed deliveries, slaughterhouse vehicles and catching team, then dead bird pick-up, egg transport (hatching eggs and table eggs), manure management and shared equipment. In terms of human movements, the most studied populations were poultry companies' technicians and/or workers, veterinarians and traders. Some references also considered vaccination and cleaning and disinfection teams [49] or private individual technicians [70].

**Table 3. Description of the results of the 48 selected references which studied animal, human and fomite movements within poultry production and trade networks.**

| Type of movements | Species considered | References |
|---|---|---|
| **International movements** | | |
| **International trade of live birds** | day-old chicks | [28,29] |
| | hatching eggs | [28] |
| | chickens | [30,31] |
| | ducks, turkeys | [31] |
| **Illegal trade of live birds** | spent hens, ducklings, day-old chicks | [27] |
| **National movements** | | |
| **Animal movements** | | |
| **Movements in link with the trade network** | poultry | [32–40,57,59–61,63,67] |
| | broiler chickens | [64,65,68–70] |
| | layers | [65,68,69] |
| | indigenous chickens | [65,68,69] |
| | ducks | [62,64,69] |
| **Between farms movements** | ostriches | [53] |
| | ducks | [44,58] |
| **Movements from hatchery** | day old chicks, hatching eggs, broiler chickens | [43,54,64–66,68,70,72,73] |
| | day old chick, hatching egg layer | [65,68,72,73] |
| | day old duckling, hatching eggs duck | [72,73] |
| | ostrich chicks | [72] |
| **Movements to poultry show** | chicken, duck | [72] |
| **Fomite movements** | | |
| **Slaughterhouses vehicles** | ducks | [41] |
| | poultry | [42,50,51,71] |
| | broiler chickens | [47,70,74] |
| | layers | [47,74] |
| **Catching team or bird pick up** | poultry | [42,48,71] |
| | broiler chickens | [46,47,55,56,74] |
| | layers | [46,47,55,56,74] |
| | turkeys and broiler ducks | [46] |
| | ready to lay parents and parents and grandparents stock | [46] |
| **Feed deliveries** | poultry | [50,52,71] |
| | broiler chickens | [43,49,54,65,68,70,73,74] |
| | layers | [49,65,68,73,74] |
| | turkeys | [49,73,74] |
| | quails and breeders | [49] |
| **Egg transport (egg tray, egg pallet, egg collection)** | layers | [45,46,55,56] |
| | parents and grandparents stock | [46] |
| **Litter and manure management** | poultry, ready to lay parents and parents and grandparents stock | [46] |
| | broiler chickens | [43,54,56,70] |
| | layers | [56] |
| **Dead birds pick-up** | broiler chickens | [56,73] |
| | layers | [72,73] |
| | ducks | [72,73] |
| | turkeys | [73] |

(*Continued*)

**Table 3.** (Continued)

| Type of movements | Species considered | References |
|---|---|---|
| **Shared equipment** | free-range layers | [56] |
| | broiler chickens | [56,70] |
| **Human movements** | | |
| **Traders** | poultry | [32,33,67] |
| **Veterinarian** | poultry | [52] |
| | broiler chickens | [65] |
| | layers | [65] |
| | ready to lay parents and parents and grandparents stock | [46] |
| **Vaccination team** | broiler chickens, layers, turkeys, quails and breeders | [49] |
| **Cleaning and disinfection team** | broiler chickens, layers, turkeys, quails and breeders | [49] |
| **Individual technician** | broiler chickens | [70] |
| **Technician/ company workers** | ducks | [41,72] |
| | poultry | [42,50] |
| | broiler chickens | [43,49,54,55,72] |
| | turkeys, quails and breeders | [49] |
| | layers | [49,55,56,72] |
| **Shared farm workers (part-time, hired help)** | broiler | [43,54,56] |
| | layers | [56] |
| **Non-company commercial services (gas delivery, meter reading, maintenance)** | broiler | [43,54] |
| **Visiting poultry show** | chickens, ducks | [72] |

## Link between animal and human movements within poultry production and trade networks and the spread of avian influenza viruses at the national and international level

From the 48 references, only 28 studied the link between animal and human movements and AI spread within poultry production and trade networks. These references were classified according to the type of network studied: international trade network (3/28), illegal trade network (1/28), national trade network (8/28) and national production network (16/28) (Table 2). None of the references on poultry production network considered the full network (i.e. from hatchery to slaughter). The references on poultry production networks were all conducted in high-income countries (HIC) while those on national poultry trade networks were all conducted in low- and middle-income countries (LMIC). Moreover, the references on trade networks all took place in Asia except for one reference (Pacific Islands [36]).

**Animal movements at the international level.**   While looking at trade at the international level, one study in the Southeast Asia region showed that the risk of a country being infected increases with the number of live chickens imported [30]. But the model of this study showed no significant interaction between infection and duck importation and a negative significant interaction between infection and turkey importation. On the other hand, a risk assessment of the introduction of HPAI H5N1 infection into Spain showed that the import of ducks was more at risk than the import of turkeys and chickens [31]. Olive et al. considered the risk of HPAI introduction in Ethiopia through DOC as negligible [29]. Looking at the risk of HPAI introduction through illegal trade in Vietnam, Desvaux et al. showed that the live poultry trade represents a high risk of HPAI introduction and considered that the risk of exposure of chickens in Vietnam was lower after the introduction of DOC than spent hens or ducklings [27].

**Animal movements within the poultry trade network at the national level.** Only one study showed an association between the movements of live poultry and the persistence of AI viruses at the farm level [36]. The seven others showed the association between live bird markets and animal movements within trade networks and risk of AI spread. Some studies showed a significant association between AI infections in live bird markets or in the market location and the movement of live birds [39,40]. Roche et al. showed the high probability of infection of a live bird market after movement of live birds from an infected village [35]. Moreover, Fournié et al. showed that live bird markets with high connectivity in a trade network had a higher probability of being infected and of infecting other markets [33] and that control measures on live bird markets such as market closure or rest day reduced the number of secondary cases per single infected case [34]. Furthermore Soares Magalhaes et al. showed that an increase of poultry trade (e.g. for Chinese New Year Festivities) influences the risk of poultry and human AI infections [38]. Live bird markets are indeed central in the trade networks in most Asian countries. Nonetheless, Martin et al. identified no significant association between movements within networks and the infectious status of the live bird market in South China [37]. It should however be noted that in this study the live bird market from counties with previous HPAI infection history was significantly less connected than the live bird market from counties with no infection history.

**Animal, fomite and human movements within the poultry production network at the national level.** At the poultry production network level, three studies conducted an analysis of movements at almost all production stages. All studies were at national level: one on fattening duck production network in France [44], another on poultry production network in the Netherlands [46] and one on the ostrich production network in South Africa [53]. Moreover, only two studies also considered breeding stocks [46,49] and one hatching egg movements [45].

Two studies identified company integration as a pathway for high risk of between-farm AI transmission [45,54]. Furthermore, two modelling studies established that the wide spread of AI within poultry production networks is possible, even if the event was rare [42,50].

Two references studied the role of animal movements within poultry production networks in AI spread. In networks with a lot of between-farm animal movements (e.g. ostriches, fattening ducks) infected farms were more central and more connected within their network as compared to the non-infected farms [44,53].

Thirteen references studied the role of fomite movements in AI spread within poultry production networks. Several studies demonstrated the role of slaughterhouse trucks in the modelled outbreak size [41,42,50]. Moreover, modelling studies concluded that the implementation of control measures on this transmission pathway reduced AI spread [47,51]. Moreover, the role of catching companies or bird pick up networks in the increase of AI spread risk has been identified [42,47,48,55,56]. Feed deliveries were identified by modelling studies as an important transmission modality [43,49,50]. Egg transport (including egg tray, egg pallet and egg collection) was identified as an important disease pathway between layer farms based on risk assessment [55,56]. Lastly, one study showed that dead bird pick-up can be considered to be an essential pathway of AI spread and that shared equipment is a possible pathway [56].

Nine references studied the role of human movements within poultry production networks in AI spread, especially company technicians or workers. Risk assessment studies highlighted these movements as an important AI risk pathway [55,56]. Modelling studies demonstrated an association between AI spread and company workers movements [41–43]. Leibler at al. identified part-time workers as significantly contributing to the increase of AI spread [54]. Movements of veterinarians along with manure, egg transport and catching team movements also

**Table 4. Risk characterisation of AI spread through the different routes within national poultry production networks identified by this literature review.**

| Type of movement | AI spread routes | Risk level | References |
|---|---|---|---|
| **Live animal movements** | Production process which requires live bird movements between farms such as fattening duck or ostrich productions | High | [44,53] |
| | Production process based on all-in/all-out system such as broiler or layer productions | Low | [43,45,46,49,54] |
| | Chick movements from hatchery | Moderate | [43,54] |
| **Fomite movements** | Bird pick-up to slaughter for broiler production | High | [46,56] |
| | Feed delivery for broiler production | High | [43,54] |
| | Egg collection for layer production | High | [45,56] |
| | Manure and litter management | Low to moderate | [43,46,54,56] |
| | Shared equipment | Moderate | [56] |
| **Human movements** | Integrated company personnel (manager, staff working on multiple premises, veterinarian) | High | [42,50] |
| | Human movements associated within in-house contact (company personnel, veterinarian, farm workers) | High | [43,45,46,54,56] |

play a role in AI spread [46]. Infected farms are more central in networks connected by farms using the same medicine business and the same feed business than non-infected farms [52].

Some references identified multiple transmission routes in AI spread within poultry production networks. Live bird movements were considered as a major route of AI spread when production processes requires live birds movements between farms (e.g. fattening duck or ostrich) [44,53]. But these movements were considered as a minor route when production processes are based on all-in/all-out system (e.g. broiler or layer) [43,45,46,49,54]. While most of the studies agreed on the other transmission routes to consider (Table 3), the relative importance of these routes with AI spread differed (Table 4). Sharkey et al. identified that more AI infections were attributable to integrated company personnel contacts than feed or slaughter contacts [50]. Dent et al. showed that the highest proportion of outbreaks occurred with integrated company personnel movements [42]. For layer production, the transmission route most at risk were fomites movements related to egg transport (e.g. egg tray and egg pallets) compared to other fomite movements (e.g. feed delivery) or human movements (e.g. company technician, veterinarian, shared farm workers) [45,56]. For broiler production, the results were more contrasted. Some studies demonstrated that feed delivery and human movements (e.g. company technician, veterinarian, farm workers) were the routes most at risk [43,54], other studies identified shared bird pick-up transport (to slaughter) as the most at risk route for AI spread [46,56].

## Discussion

First, this review highlighted the fact that only a limited number of studies looked at the role of animal, human and fomite movements within poultry production and trade networks in the spread of AI. All studies describing a link between these movements and AI spread were actually based on modelling work (e.g. epidemiological, network analysis or risk assessment models) rather than experimental work. Indeed, it is highly challenging to show evidence of a link between a specific movement and the cause of an outbreak in the field, given the complexity and numbers of movements involved. Studies using pre-movement samplings can only show the presence of the virus but cannot prove that this virus will infect another bird once the movement has occurred [58]. Most studies are based on modelling and would require field validation to improve our global knowledge on this issue. Moreover, only a few studies used real AI outbreak data to parameterize the models [37–40,44,52,53].

The role of human and fomite movements has mostly been studied within national poultry production networks but never within international poultry production networks (e.g. poultry production company with production units in several countries). This role should not be neglected. At the national poultry production network level, studies have shown that human and fomite movements were more important routes than live bird movements [43,45,46, 49,54]. Indeed, even if the transmission probability through direct contact from one infected bird to another is the highest of all transmission routes, the probability of this contact occurring is low. This is due to the poultry production process with the all-in and all-out system. Live birds leaving a farm to go to slaughter ends the transmission risk [45,49]. This process does not apply to fattening duck and ostrich production networks where live bird movements play a large role in AI transmission [44,53]. At the national poultry trade level, most studies made no distinction between traders' movements and the movements of the poultry that they trade. It is therefore impossible to identify the link between human and fomite movements and AI spread in trade networks, even if this link certainly exists, as for the production network. Some articles suggested that the global trade of live domestic birds or poultry by-products may play a major role in the global spread of AI [28,84,85]. Only a few references actually demonstrated this role and mainly linked to live animal trade networks [27,29–31]. Most of these studies used risk assessment and this literature review may have missed other risk assessment studies at national or local level due to selecting English written papers only. It would be interesting to look at this work led by national veterinary services. Nonetheless, as these risk assessment studies focused at national level, the studies do not usually address the link between movements and AI spread at global level.

Considering the large number of references on AI in the literature, the structured selection process using PRISMA-ScR guidelines mitigated the risk of missing key articles in scope with the research questions. Only two references were identified through subsequent citations, which highlighted the strength of the search algorithm. The main target of this review was international and national poultry production and trade networks. However, as farms are an element of the poultry production and trade networks, many studies initially identified by the algorithm (and removed with the exclusion criteria) targeted AI spread at farm level only (within farms or between farms) but not in direct link with the whole network.

Even if a majority of references considered only HPAI spread, this review also looked at papers focused on LPAI spread [41] or both LP and HP spread [55,56,82]. Indeed, transmission pathways are highly similar for HPAI and LPAI, even if transmission probability might vary [55]. Moreover, LPAI are more likely to spread widely than HPAI due to the lack of detection and reporting [55,56]. Furthermore, Nickbakhsh et al. have shown that LPAI and HPAI co-infections can lead to a prolongation of HPAI outbreak due to partial cross-protection [82]. LPAI might therefore play a critical role in HPAI surveillance and control performances.

It is also important to note that no references on full production networks (from hatchery to slaughter) were identified from this review, even if some references looked at movements at several stages of the whole poultry production network [44,46,53]. One reason could be the lack of data at some stages of the production network (e.g. hatchery or slaughterhouse). In addition, when they do exist, data from the private sector are confidential and the veterinary services and researchers have limited access to them. Similarly, data on poultry production networks at the international level are not publically accessible essentially due to confidentiality issues from private producers [28]. Therefore, all studies rely on public official data, which are often outdated and of lower precision and quality than private sector data. Access to these data is key for a full understanding of the role of movements within poultry production and trade networks in the global spread of AI. Stronger public-private partnerships could facilitate public data access and improve disease control benefits for the private sector [86,87].

It is interesting to note that most studies on poultry production networks have been conducted in HIC while all studies on trade networks have been conducted in LMIC. On one hand, the structure of poultry production networks in LMIC with many small-scale farms needs live domestic bird trade networks to structure this network. As the biosecurity level on these small-scale farms is low, they represent a high risk of AI spread in the country. This could explain why studies conducted in LMIC mostly focused on trade networks. On the other hand, the trade of live domestic birds in HIC has become marginal compared to the trade of poultry by-products and poultry production is highly integrated, hence the final product of the value-chain for the consumers is poultry meat. Poultry trade networks have been well studied in LMIC, especially during the 2004–2006 H5N1 pandemic waves, and their role in disseminating AI viruses has been evidenced. Live bird markets have been identified as a major source of human infections [88]. Poultry production networks including international ones are present in LMIC and sharply connected to traditional farming (backyard) and live bird trade [89]. Only a few reports described the poultry value-chain in developing countries with high poultry production, in particular Egypt [90] and Indonesia [69,91] and with mid-size poultry production, i.e. Kenya [65,92] and Nepal [68]. Yet, this description was only intended to provide information for biosecurity, surveillance and control measures rather than trying to understand the role of animal and human mobility within these networks in the spread of AI. Limited data access and possibility of intervention could explain why studies led by international organisations have focused on backyard, semi-commercial and trade networks, including live bird markets.

At the international level, live poultry and the poultry by-product trade is controlled by trade regulations issued by the OIE [93]. If a country wants to export live poultry and poultry by-products, its national regulatory authorities have to provide evidence that the country or zone/compartment within the country is free of AI. For this, they should use active and passive surveillance, according to the recommendations of the OIE Terrestrial Code 2018 [93]. AI outbreaks have huge consequences on international trade; even a short-term crisis can have a long term impact on trading patterns and policy decisions as well as on industry development [94]. For example, following the 2003–2004 HPAI H5N1 epidemic in South East Asia, Japan stopped importing frozen poultry meat from China and Thailand and increased its importations from Brazil [95]. The economic impact of the 2003–2004 H5N1 AI wave in Thailand due to trade bans was estimated at almost 1.5% of gross domestic product [96]. Taking the US as another example, trade bans imposed by trading partners after the 2014–2015 HPAI outbreaks were valued at almost 14% of the year's total trade revenue [97]. Nevertheless, the potential role of trade in the risk of global spread of animal diseases has been discussed but not extensively studied, as shown in this review work.

The role of wild birds in the dissemination of AI viruses has been extensively studied [10]. Phylogenetic epidemiology can provide clues to identify the source of introduction and spread modalities of AI [7,98]. Nonetheless, in some cases, the initial source of introduction of the virus has never been clearly evidenced, especially between infected wild birds or contaminated movements within international poultry trade and production networks. This work has highlighted the fact that, even if wild birds were contributing to AI spread and maintenance at the global level, other factors including animal, human and fomite movements within poultry production and trade networks play a role in the global spread of AI. Although every context is different depending on the countries considered, it would be interesting to run a comparative risk analysis on the risk of introduction and spread of AI from wild birds or within poultry production and trade network movements at the global level.

## Conclusion

Despite the intensive circulation of AI in the last 15 years and the large number of studies that have examined its spread, this review has shown that only a limited number of studies have been focused on the role played by animal, human and fomite movements in this spread within poultry production and trade networks. Most studies have described the different AI spread routes, but without looking at the risk associated with these routes, especially between commercial farms and the rest of the production and trade networks. This work has confirmed that production and trade networks are considered to play a role in AI spread but they have only been studied partially. Animal movements play an important role in AI spread within national trade networks whereas human and fomite movements play an important role in AI spread within national poultry production networks. Nevertheless, this has never been demonstrated with field data. Although, the international legal and illegal trade of live poultry is recognised as possible routes of AI spread between countries, this has never been directly studied. A more holistic approach to the AI circulation routes including all compartments, such as commercial farms, production and trade networks, wild birds, environment and backyard, is essential to fully understand the global spread of AI and to inform relevant surveillance and control strategies. The complexity of poultry production and of AI spread is most likely part of the reasons for these gaps, but the limited access to production and trade data for public researchers is definitely a barrier to this knowledge. There is therefore a need to study the role of animal, human and fomite movements within poultry production and trade networks in the global spread of AI in partnership with both public and private actors to fill this gap.

## Supporting information

**S1 Table. Preferred Reporting Items for Systematic Reviews and Meta-Analyses extension for Scoping Reviews (PRISMA-ScR) checklist.** NA: not applicable.
(PDF)

**S2 Table. List of the different strings included in the search strategy and the number of retrieved references for each string.** Application on Scopus database on the 31 May 2019.
(PDF)

**S3 Table. Listing of the objectives, type, methods and main results of the 57 selected references.** UK: United Kingdom, USA: United States of America.
(PDF)

## Acknowledgments

The authors acknowledge the reviewers for their valuable comments, which have greatly helped them to improve the manuscript.

## Author Contributions

**Conceptualization:** Claire Hautefeuille, Gwenaëlle Dauphin, Marisa Peyre.

**Data curation:** Claire Hautefeuille.

**Formal analysis:** Claire Hautefeuille.

**Funding acquisition:** Gwenaëlle Dauphin, Marisa Peyre.

**Investigation:** Claire Hautefeuille.

**Methodology:** Claire Hautefeuille, Marisa Peyre.

**Resources:** Marisa Peyre.

**Supervision:** Gwenaëlle Dauphin.

**Visualization:** Claire Hautefeuille.

**Writing – original draft:** Claire Hautefeuille.

**Writing – review & editing:** Gwenaëlle Dauphin, Marisa Peyre.

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
