## [Decision Letter · Decision Letter 0]

16 Oct 2019

PONE-D-19-20368

Global spread of avian influenza viruses: the role of animal and human movements in poultry production and trade networks – a scoping review

PLOS ONE

Dear Dr Hautefeuille,

Thank you for submitting your manuscript to PLOS ONE. After careful consideration, we feel that it has merit but does not fully meet PLOS ONE’s publication criteria as it currently stands. Therefore, we invite you to submit a revised version of the manuscript that addresses the points raised during the review process.

We would appreciate receiving your revised manuscript by Nov 30 2019 11:59PM. To enhance the reproducibility of your results, we recommend that if applicable you deposit your laboratory protocols in protocols.io, where a protocol can be assigned its own identifier (DOI) such that it can be cited independently in the future. For instructions see: http://journals.plos.org/plosone/s/submission-guidelines#loc-laboratory-protocols

We look forward to receiving your revised manuscript.

Kind regards,

Maria Serena Beato

Academic Editor

PLOS ONE

Journal Requirements:

This study was funded by Ceva Santé Animale (https://www.ceva.com/en/). The funders had no role in study design, data collection and analysis, decision to publish, or preparation of the manuscript.

We note that you received funding from a commercial source: Ceva Santé Animale

Additional Editor Comments:

I would like to request to pay careful attention to each comment of each referee in order to

Improve substantially the work presented. Issues in the methodology have been raised by both referees. In addition a better explanation of the aims of the review presented is necessary.

Reviewers' comments:

Reviewer's Responses to Questions

**Comments to the Author**

1. Is the manuscript technically sound, and do the data support the conclusions?

Reviewer #1: Yes

Reviewer #2: Partly

2. Has the statistical analysis been performed appropriately and rigorously? 

Reviewer #1: N/A

Reviewer #2: N/A

3. Have the authors made all data underlying the findings in their manuscript fully available?

Reviewer #1: Yes

Reviewer #2: Yes

4. Is the manuscript presented in an intelligible fashion and written in standard English?

Reviewer #1: No

Reviewer #2: Yes

5. Review Comments to the Author

Reviewer #1: Comments and suggestions to the author(s):

In this study, Hautefeuille et al. conducted a scoping review according to the PRISMA guidelines extended for Scoping Reviews, with the primary goal of identifying gaps in the literature concerning the role of animal and human movements within the poultry production and trade networks that contribute to the global spread of avian influenza virus. The authors examined the extent, range and nature of the evidence on this topic, summarizing findings form a body of knowledge encompassing heterogeneous sources. As avian influenza continues posing threats to animal and human health, such a scoping review is of particular interest to researchers and policymakers as it may aid the planning and commissioning of future research.

However, some issues need to be properly addressed and discussed by the authors.

Major concerns

The manuscript needs to be professionally proofread. Incomplete and/or long sentences and concepts, poor grammar or punctuation detract from readability and hamper the scientific merit and originality of this manuscript, so it needs major rewriting for sense and flow. The grammar and English revision is beyond my task, therefore I can only provide few examples along the manuscript that were particularly hard to understand:

Page 1 lines1-2: the title itself to begin with; as it is stated it is not clear that this manuscript is a scoping review aiming at mapping evidence/identifying main concepts, theories, sources, and knowledge gaps on the role of animal and human movements within the poultry production and trade networks in the global spread of AI viruses.

Page1-2 lines 33-40: two sentences repeating the same concepts, but remaining vague on the conclusion. It is clear that there is a general awareness of the role of animal and human movements in the local and global spread of AI, but it is not clear what the authors identified as gap(s) of knowledge: they only mention ‘fine understanding’ in line 35 and ‘limited work has been carried out to study it’ on lines 38-40. Do they mean that no study quantified the proportion of AI virus transmission/spread due to animal and/or human movements within poultry production and trade networks? Do they mean that no study identified which is the most important route(s) of transmission/spread of AI within the production and trade networks? It would be worthy to elaborate a more precise, though concise sentence, on this issue in the abstract and then to elaborate on the concept throughout the manuscript, when commenting the groups of studies selected. For example on page 11 lines 222-224 the sentence is not clear enough. What the authors refer to when they say ‘this study’? Which of the studies mentioned in the paragraph lines 215-224.

Page 7 lines 135-137: ‘on title, abstract and full text if necessary using the following exclusion criteria on title, abstract and full text if necessary’, two repetitions in one sentence.

Page 9 lines 180-182: this sentence is not clear, and it seems incomplete. Please rephrase.

Page 9 line 192: in the figure 2 title, it says ‘number without comma’, but there is no number with or without comma in the figure itself. Please clarify.

Page 8 line 167; page 11 line 211: Spell out numbers (e.g., one, two, three) at the start of sentences rather than using numerals.

Page 10 line 205: in the title of Table 1, please explain the colour code used in the table.

Moreover, it would help to clarify the following concepts and passages in the text:

In the Introduction, please be more specific with respect to the years considered in the review: explain the reason why AI outbreaks were considered only starting from 2010 (page 3 lines 45; line 55; line 61), although the search strategy included years from 1975 to 2019 (page 5 line 104-105).

Material and methods: it would help to find the same section structure as provided in the checklist/S1 Table and that is: protocol & registration; eligibility criteria; information sources ect… Those information are all present in the text but not is a structured manner, which is also one of the principles of the PRISMA evidence-based system for reporting items in reviews.

Results: unclear, but key components are present and major rewriting can address lack of clarity

Page 11 lines 209 onwards: In the results it would help to find the same structure outlined in Figure 1 (Flow chart diagram), so that the details concerning the references could be discussed in three different sections reflecting the three groups/boxes printed in Fig. 1. As it is discussed now, it is not clear enough.

Figure 1: please be more specific in the definition assigned to the three final groups of references selected, which do not correspond to what is reported in the text (page 11 lines 216-220)

-First box, 28 references on the role of animal and human movements […]: what do you mean by ‘the role’? At page 11 line 217, those 28 references are referred to as corresponding to references on the link between animal and human movements, which are the references in the second box, as reported in Figure 1;

-Second box, 20 animal and human movements in link with […]: what does it mean ‘in link with AI spread’? The entire review is about literature on animal and human movements identified within the poultry production and trade network that are linked to AI spread.

-Third box, 10 references on AI spread within poultry production and trade network […]: are those references specifically quantifying the circulation of AI viruses only within these environments and transmission/spreading mechanisms specifically ascribed to animal and human movements?

Discussion: it lacks of structure. The Discussion should start with one or two sentences that prepare the ground for making considerations on the results obtained. Instead, the authors, after the first sentence about the structured methodology used, started with listing the major limitations or risks and how they were dealt with. Gaps in knowledge are stated only on page 26 lines 399-400.

Page 28 lines 460-463: one reference on risk assessment conducted during an endemic period against many other risk assessment studies conducted in non epidemic periods, does not seem to be a solid ground on which drawing such a conclusion. Please elaborate more on this idea or provide more details on the references and their perspective.

Conclusions: page 30 lines 497-501, repetitions of concepts. Please make one sentence out of the concept repeated concerning the fact that the role of animal and human movements within poultry production and trade networks is acknowledged but poorly studied. Moreover, please elaborate on which aspect(s) is/are poorly studied and can be topic for future research.

Reviewer #2: The article “global spread of avian influenza viruses: the role of animal and human movements in poultry production and trade networks – a scoping review” is a scoping review. The study seeks to understand better the role played by the movement of animals and people a) within the poultry production network and b) the trading network, on the global spread of avian influenza. The authors declare that the methodology of this review follows the PRISMA guidelines. The authors a) report on risk factors of AI in articles identified through the search strategy; b) classify articles according to the type of study, aim, and location; and c) list international and national movements of animals, fomites and people in studies that were selected for the review.

For this review, I declare knowledge in systematic review methodology, poultry production and avian influenza.

The authors must note that their submission is a “scoping review”. According to PRISMA, scoping reviews help to “determine whether a systematic review of the literature is warranted”. The authors declared to have followed the PRISMA guidelines (for systematic reviews) in the main text, but the supplementary material 1 reports on PRISMA ScR – an extension of such guidelines to conduct scoping reviews. I will provide general feedback on the manuscript and specific feedback based on the PRISMA ScR checklist.

The manuscript covers a relevant topic given the ongoing reporting of AI outbreaks in different parts of the globe. However, the study requires substantial revision. I recommend the authors a full revision of the manuscript to meet PRISMA guidelines.

Introduction and Abstract: There are significant issues with the objective of this study. In the abstract, the objective is to increase understanding of the role of animal trade. In the introduction, the objective is to understand the role of animal and human movement within the poultry production network, missing background on the animal trading network. In the results section, authors report on the movement of animals, humans AND fomites; fomites were not declared as part of the aims of the study in the abstract or introduction. Also, the abstract requires a well-structured summary of the review methodology.

Methodology: PRISMA items that require attention are marked in the supplementary material 1. Overall, multiple items of the PRISMA ScR checklist are missing. There isn't a research question and there isn't an explicit step-by-step protocol. A good protocol must describe the decision process, the exclusion criteria, methods to address difference of opinion, and inform the authors involved in each step, among others.

A research question and a clear protocol are essential to guide the review. To demonstrate this, I want to highlight the articles removed by the exclusion criterion “references focused on risk factors of AI spread”. Although removed, these articles were analysed and results were presented. These analyses were not declared in the study aim. Also, Figure 1 shows that only 58 papers made it to the final selection to be analysed and reported.

Results: It is difficult to evaluate these results in the absence of a clear research question. Based on the aims declared in the title, abstract and introduction only, the reporting of risk factors and movement of fomites do not align with the objectives of this paper. However, these results may be incredibly relevant to the right research questions. Again, in manuscript line 178, the authors report on routes of avian influenza spread (in a total of 276 articles) which was not declared in the protocols, the aims of the study, or in Figure 1. Table 2 classify and characterises the articles selected, but a better synthesis of the results that emerge from grouping the selected articles in three silos is expected.

Conclusion: The conclusion of this article will improve when a clear research question exists. The authors do not conclude or tell us what is known; therefore, we do not improve our understanding of this topic. For instance, what is riskier? The movement of veterinarians or that of vaccinators? Or, what species movement is the riskiest based on modelling, risk assessment and other techniques? This review can respond to this kind of questions.

Specific comments:

Define the following terms in the manuscript: “production system”, “production network”, “production trade”, “trading network”, “compartment”. If possible, remove some of these terms from the manuscript and use the remaining consistently.

Line 93: Declare PRISMA ScR guidelines if this is a scoping review.

Line 185-186 “only a limited number of references studied the role of international trade” – the authors included the word “domestic” but not “international in their search terms.

Table 1. risk factor from wild birds to backyard marked NA: based on the Asian crisis of H5N1, wild birds and backyard free-ranging chickens represented a high-risk factor for AI spread. I would expect to see a Zero in that cell if this risk factor was not mentioned in any article. Table 1needs to explain what NA and zero represent.

Lines 221-223: why is that clarification necessary?

Table 2: papers that evaluate the risk of introduction and dissemination. Systematic (or scoping review) protocol needs to make explicit why risk assessment is eligible and why risk factors are not. For instance, in line 359 the authors refer to movements as important “AI risk pathways”, which may be understood as certain activities being “risk factors”, and risk factors are excluded from this review. Also, the authors must discuss how a risk assessment demonstrates links between animal and human movement and AI spread, particularly when the risk associated with movement in the original studies is deemed negligible.

Table 2: Studies on AI spread only (referring to animal and human movements): The sentence in parenthesis was not explained in the main text. This sentence is not clear.

Line 243 (and others): instead of “vet” use “veterinary.”

Line 373: Discussion highlights the exclusion phases to achieve a selection of 58 papers split into three groups, yet the authors reported on excluded papers in their result section. This inconsistency can be amended with an adequate research question and protocols.

Line 386-389: what are the implications for this review?

Lines 432-462: Low- and middle-income countries (LMIC) are expected to have small-scale production of poultry, while industrialised countries to have large production systems. Also, LMIC systems have less biosecurity which may translate into a higher risk of having an enzootic circulation of AI. In this context, it makes sense that most research on trading networks is directed to systems in LMIC, while the investigation of poultry production networks is more focused on industrialised countries. I suggest emphasising on the discussion of contexts that explain this dichotomy.

6. PLOS authors have the option to publish the peer review history of their article (what does this mean?). If published, this will include your full peer review and any attached files.

Reviewer #1: No

Reviewer #2: No

---

## [Author Response · Author response to Decision Letter 0]

29 Nov 2019

Journal Requirements:

The manuscript was corrected to meet PLOS ONE’s style requirements, including the file naming.

This study was funded by Ceva Santé Animale (https://www.ceva.com/en/). The funders had no role in study design, data collection and analysis, decision to publish, or preparation of the manuscript.

We note that you received funding from a commercial source: Ceva Santé Animale

An amended Competing Interests Statement was included in the cover letter.

Additional Editor Comments:

I would like to request to pay careful attention to each comment of each referee in order to

Improve substantially the work presented. Issues in the methodology have been raised by both referees. In addition a better explanation of the aims of the review presented is necessary.

Thank you for your comment. We considered all reviewers’ comments, which have greatly helped us to improve the manuscript. The aims of the review and the methodology section were improved according to these comments.

Reviewers' comments:

Review Comments to the Author

Reviewer #1: Comments and suggestions to the author(s):

In this study, Hautefeuille et al. conducted a scoping review according to the PRISMA guidelines extended for Scoping Reviews, with the primary goal of identifying gaps in the literature concerning the role of animal and human movements within the poultry production and trade networks that contribute to the global spread of avian influenza virus. The authors examined the extent, range and nature of the evidence on this topic, summarizing findings form a body of knowledge encompassing heterogeneous sources. As avian influenza continues posing threats to animal and human health, such a scoping review is of particular interest to researchers and policymakers as it may aid the planning and commissioning of future research.

However, some issues need to be properly addressed and discussed by the authors.

Major concerns

The manuscript needs to be professionally proofread. Incomplete and/or long sentences and concepts, poor grammar or punctuation detract from readability and hamper the scientific merit and originality of this manuscript, so it needs major rewriting for sense and flow. The grammar and English revision is beyond my task, therefore I can only provide few examples along the manuscript that were particularly hard to understand:

The paper has been revised and corrected by a professional translator specialise in scientific editing. 

Page 1 lines1-2: the title itself to begin with; as it is stated it is not clear that this manuscript is a scoping review aiming at mapping evidence/identifying main concepts, theories, sources, and knowledge gaps on the role of animal and human movements within the poultry production and trade networks in the global spread of AI viruses.

The title has been amended to better reflect the objectives of this scoping review.

Page1-2 lines 33-40: two sentences repeating the same concepts, but remaining vague on the conclusion. It is clear that there is a general awareness of the role of animal and human movements in the local and global spread of AI, but it is not clear what the authors identified as gap(s) of knowledge: they only mention ‘fine understanding’ in line 35 and ‘limited work has been carried out to study it’ on lines 38-40. Do they mean that no study quantified the proportion of AI virus transmission/spread due to animal and/or human movements within poultry production and trade networks? Do they mean that no study identified which is the most important route(s) of transmission/spread of AI within the production and trade networks? It would be worthy to elaborate a more precise, though concise sentence, on this issue in the abstract and then to elaborate on the concept throughout the manuscript, when commenting the groups of studies selected. 

The unclear sentences were modified. The results and discussion section were amended to present the current knowledge and gaps (what has been demonstrated, what has been suggested but not demonstrated and what has not been studied at all) on the risk link with animal, human and fomite movements. This is now reflected back in the revised version of the abstract.

For example on page 11 lines 222-224 the sentence is not clear enough. What the authors refer to when they say ‘this study’? Which of the studies mentioned in the paragraph lines 215-224.

This sentence was confusing and therefore was removed.

Page 7 lines 135-137: ‘on title, abstract and full text if necessary using the following exclusion criteria on title, abstract and full text if necessary’, two repetitions in one sentence.

Corrected

Page 9 lines 180-182: this sentence is not clear, and it seems incomplete. Please rephrase.

Corrected

Page 9 line 192: in the figure 2 title, it says ‘number without comma’, but there is no number with or without comma in the figure itself. Please clarify.

Corrected

Page 8 line 167; page 11 line 211: Spell out numbers (e.g., one, two, three) at the start of sentences rather than using numerals.

Corrected

Page 10 line 205: in the title of Table 1, please explain the colour code used in the table.

Corrected

Moreover, it would help to clarify the following concepts and passages in the text:

In the Introduction, please be more specific with respect to the years considered in the review: explain the reason why AI outbreaks were considered only starting from 2010 (page 3 lines 45; line 55; line 61), although the search strategy included years from 1975 to 2019 (page 5 line 104-105).

A reference to AI outbreaks identification at the end of the nineteen century was added in the introduction to account for all outbreaks since the first identification of the disease. The search strategy was from 1975 because the oldest references listed in literature search databases were from 1975.

Material and methods: it would help to find the same section structure as provided in the checklist/S1 Table and that is: protocol & registration; eligibility criteria; information sources ect… Those information are all present in the text but not is a structured manner, which is also one of the principles of the PRISMA evidence-based system for reporting items in reviews.

The structure of the methodology section has been adapted to follow the PRISMA ScR checklist as the reviewer recommended.

Results: unclear, but key components are present and major rewriting can address lack of clarity

Page 11 lines 209 onwards: In the results it would help to find the same structure outlined in Figure 1 (Flow chart diagram), so that the details concerning the references could be discussed in three different sections reflecting the three groups/boxes printed in Fig. 1. As it is discussed now, it is not clear enough.

Figure 1: please be more specific in the definition assigned to the three final groups of references selected, which do not correspond to what is reported in the text (page 11 lines 216-220)

Thank you for this comment. The three groups’ terminology was amended to improve clarity of the wording. The same terminology was used for the three groups all along the manuscript and in the Fig 1.

-First box, 28 references on the role of animal and human movements […]: what do you mean by ‘the role’? At page 11 line 217, those 28 references are referred to as corresponding to references on the link between animal and human movements, which are the references in the second box, as reported in Figure 1;

-Second box, 20 animal and human movements in link with […]: what does it mean ‘in link with AI spread’? The entire review is about literature on animal and human movements identified within the poultry production and trade network that are linked to AI spread.

-Third box, 10 references on AI spread within poultry production and trade network […]: are those references specifically quantifying the circulation of AI viruses only within these environments and transmission/spreading mechanisms specifically ascribed to animal and human movements?

The reason for the split between these three groups and the meaning of the chosen terminology was better explained in the results section of the manuscript. This should address the three points requested above.

Discussion: it lacks of structure. The Discussion should start with one or two sentences that prepare the ground for making considerations on the results obtained. Instead, the authors, after the first sentence about the structured methodology used, started with listing the major limitations or risks and how they were dealt with. Gaps in knowledge are stated only on page 26 lines 399-400.

The structure of the discussion has been revised and improved according to this comment.

Page 28 lines 460-463: one reference on risk assessment conducted during an endemic period against many other risk assessment studies conducted in non epidemic periods, does not seem to be a solid ground on which drawing such a conclusion. Please elaborate more on this idea or provide more details on the references and their perspective.

We have chosen to remove this part to ease the discussion and clarify the main messages of this scoping review.

Conclusions: page 30 lines 497-501, repetitions of concepts. Please make one sentence out of the concept repeated concerning the fact that the role of animal and human movements within poultry production and trade networks is acknowledged but poorly studied. Moreover, please elaborate on which aspect(s) is/are poorly studied and can be topic for future research.

The repetition has been deleted. Additional sentences on the aspects which have been poorly studied and which could be topic for future research were added.

Reviewer #2: The article “global spread of avian influenza viruses: the role of animal and human movements in poultry production and trade networks – a scoping review” is a scoping review. The study seeks to understand better the role played by the movement of animals and people a) within the poultry production network and b) the trading network, on the global spread of avian influenza. The authors declare that the methodology of this review follows the PRISMA guidelines. The authors a) report on risk factors of AI in articles identified through the search strategy; b) classify articles according to the type of study, aim, and location; and c) list international and national movements of animals, fomites and people in studies that were selected for the review.

For this review, I declare knowledge in systematic review methodology, poultry production and avian influenza.

The authors must note that their submission is a “scoping review”. According to PRISMA, scoping reviews help to “determine whether a systematic review of the literature is warranted”. The authors declared to have followed the PRISMA guidelines (for systematic reviews) in the main text, but the supplementary material 1 reports on PRISMA ScR – an extension of such guidelines to conduct scoping reviews. I will provide general feedback on the manuscript and specific feedback based on the PRISMA ScR checklist.

The authors have indeed followed the PRISMA ScR guidelines. This mistake was corrected in the method section of the manuscript.

The manuscript covers a relevant topic given the ongoing reporting of AI outbreaks in different parts of the globe. However, the study requires substantial revision. I recommend the authors a full revision of the manuscript to meet PRISMA guidelines.

This was done and has helped to improve the manuscript clarity, more information is detailed below.

Introduction and Abstract: There are significant issues with the objective of this study. In the abstract, the objective is to increase understanding of the role of animal trade. In the introduction, the objective is to understand the role of animal and human movement within the poultry production network, missing background on the animal trading network. In the results section, authors report on the movement of animals, humans AND fomites; fomites were not declared as part of the aims of the study in the abstract or introduction. Also, the abstract requires a well-structured summary of the review methodology.

The objectives of the study have been clarified: they now include trade networks and fomite movements. Moreover, background on animal trading network has been added. 

Methodology: PRISMA items that require attention are marked in the supplementary material 1. Overall, multiple items of the PRISMA ScR checklist are missing. There isn't a research question and there isn't an explicit step-by-step protocol. A good protocol must describe the decision process, the exclusion criteria, methods to address difference of opinion, and inform the authors involved in each step, among others.

A research question and a clear protocol are essential to guide the review. To demonstrate this, I want to highlight the articles removed by the exclusion criterion “references focused on risk factors of AI spread”. Although removed, these articles were analysed and results were presented. These analyses were not declared in the study aim. Also, Figure 1 shows that only 58 papers made it to the final selection to be analysed and reported.

Thank you for this comment. We have specified more clearly the research questions. We have now two research questions:

- What are the different routes of AI spread ? 

- What are the role of animal and human movements, including fomites, in the global spread of AI within poultry production and trade networks?

These research questions clarify the issue about analysing references focusing on risk factors to address the first one.

We have made the step by step protocol followed using the PRISMA ScR checklist more explicit in our method section.

Fig 1 was amended to account for these changes.

Results: It is difficult to evaluate these results in the absence of a clear research question. Based on the aims declared in the title, abstract and introduction only, the reporting of risk factors and movement of fomites do not align with the objectives of this paper. However, these results may be incredibly relevant to the right research questions. Again, in manuscript line 178, the authors report on routes of avian influenza spread (in a total of 276 articles) which was not declared in the protocols, the aims of the study, or in Figure 1. Table 2 classify and characterises the articles selected, but a better synthesis of the results that emerge from grouping the selected articles in three silos is expected.

The result section of the manuscript was improved with the addition of sentences associating results with research questions.

Conclusion: The conclusion of this article will improve when a clear research question exists. The authors do not conclude or tell us what is known; therefore, we do not improve our understanding of this topic. For instance, what is riskier? The movement of veterinarians or that of vaccinators? Or, what species movement is the riskiest based on modelling, risk assessment and other techniques? This review can respond to this kind of questions.

As the research questions were clarified, the conclusion was adapted to inform the reader about what is known and what is still missing. Additional results on the movement type which were most at risk in terms of AI spread was provided and discussed (also requested by the other reviewer). The conclusion and the abstract were also improved with the addition of those results.

Specific comments:

Define the following terms in the manuscript: “production system”, “production network”, “production trade”, “trading network”, “compartment”. If possible, remove some of these terms from the manuscript and use the remaining consistently.

The authors chose to use only production network and trade network for all the similar terms used with the exception of compartment, which is used only to describe the figure 2. In this case, the term compartment is used to define area in which or between which AI viruses can spread.

Line 93: Declare PRISMA ScR guidelines if this is a scoping review.

Corrected

Line 185-186 “only a limited number of references studied the role of international trade” – the authors included the word “domestic” but not “international in their search terms.

The word “domestic” was included to focus on “production bird” rather than “wild birds”. We were interested on references at the international level but also on references at all other levels (the regional, national or local level). This is the reason why no term were added in the search terms to specify for the geographical level.

Table 1. risk factor from wild birds to backyard marked NA: based on the Asian crisis of H5N1, wild birds and backyard free-ranging chickens represented a high-risk factor for AI spread. I would expect to see a Zero in that cell if this risk factor was not mentioned in any article. Table 1needs to explain what NA and zero represent.

A clarification was made in the legend of the figure. NA was used in cells linking compartments not included especially in the focus of this literature search (environment, wild birds, and backyard): the number of references retrieved would not be representative of the real number of references from the literature on this topic. 

Lines 221-223: why is that clarification necessary?

This sentence was deleted as it was not necessary.

Table 2: papers that evaluate the risk of introduction and dissemination. Systematic (or scoping review) protocol needs to make explicit why risk assessment is eligible and why risk factors are not. For instance, in line 359 the authors refer to movements as important “AI risk pathways”, which may be understood as certain activities being “risk factors”, and risk factors are excluded from this review. Also, the authors must discuss how a risk assessment demonstrates links between animal and human movement and AI spread, particularly when the risk associated with movement in the original studies is deemed negligible.

We did not make a difference between references on risk factors and on risk assessment. This was not clear in the previous version of the manuscript and led to thus above mention confusion. References on risk factors/risk assessment were removed using the exclusion criteria 3) (focus on risk factors without considering animal, human and fomite movements). We have amended the text in the method, result and discussion sections and also in Fig 1 to clarify this issue.

Table 2: Studies on AI spread only (referring to animal and human movements): The sentence in parenthesis was not explained in the main text. This sentence is not clear.

Table 2 was amended to improve clarity and reflect changes made in the groups classification.

Line 243 (and others): instead of “vet” use “veterinary.”

Corrected

Line 373: Discussion highlights the exclusion phases to achieve a selection of 58 papers split into three groups, yet the authors reported on excluded papers in their result section. This inconsistency can be amended with an adequate research question and protocols.

This inconsistency was resolved with the clarification of the research questions described above and improved methodology following PRSIMA guidelines which justify the reporting on some of the paper subsequently excluded in review.

Line 386-389: what are the implications for this review?

This part was removed as it only highlighted the difficulty in the search process but it doesn’t have any implication in the results of the review. 

Lines 432-462: Low- and middle-income countries (LMIC) are expected to have small-scale production of poultry, while industrialised countries to have large production systems. Also, LMIC systems have less biosecurity which may translate into a higher risk of having an enzootic circulation of AI. In this context, it makes sense that most research on trading networks is directed to systems in LMIC, while the investigation of poultry production networks is more focused on industrialised countries. I suggest emphasising on the discussion of contexts that explain this dichotomy.

A sentence was added to improve the context of this paragraph. Moreover, the part on the dichotomy observed in epidemiological status has been deleted, as it was not necessary for discussion.

Moreover, one of the two reviewers added the following comment within the PRISMA ScR checklist (S1_Table): “Need to present a search strategy including number of papers retrieved by each string and the total of papers retrieved from the source when strings are combined”

An extra table was added in Supporting Information (S2_Table) in order to present the number of papers retrieved by each string and the total of papers retrieved when strings are combined from one source.

---

## [Decision Letter · Decision Letter 1]

4 Mar 2020

Knowledge and remaining gaps on the role of animal and human movements in the poultry production and trade networks in the global spread of avian influenza viruses – a scoping review

PONE-D-19-20368R1

Dear Dr. Hautefeuille,

We are pleased to inform you that your manuscript has been judged scientifically suitable for publication and will be formally accepted for publication once it complies with all outstanding technical requirements.

With kind regards,

Charles J. Russell, Ph.D.

Academic Editor

PLOS ONE

Additional Editor Comments (optional): I was re-assigned this manuscript today. After browsing the original submission, considering the reviewers' comments, and reading the revised manuscript and rebuttal, I agree with the reviewer that the resubmission is substantially improved and worthy of acceptance. The writing is better than many manuscripts I have read. The review is thoughtful and comprehensive. The conclusion, that more and systemic studies are needed, is a fair statement supported by the review of the literature. Thank you for providing this review. Based on the reviewer and my review, I do not believe any further reviewer opinions were needed at this stage. I appreciate your patience.

Reviewers' comments:

Reviewer's Responses to Questions

**Comments to the Author**

1. If the authors have adequately addressed your comments raised in a previous round of review and you feel that this manuscript is now acceptable for publication, you may indicate that here to bypass the “Comments to the Author” section, enter your conflict of interest statement in the “Confidential to Editor” section, and submit your "Accept" recommendation.

Reviewer #1: All comments have been addressed

2. Is the manuscript technically sound, and do the data support the conclusions?

Reviewer #1: Yes

3. Has the statistical analysis been performed appropriately and rigorously? 

Reviewer #1: N/A

4. Have the authors made all data underlying the findings in their manuscript fully available?

Reviewer #1: Yes

5. Is the manuscript presented in an intelligible fashion and written in standard English?

Reviewer #1: Yes

6. Review Comments to the Author

Reviewer #1: (No Response)

7. PLOS authors have the option to publish the peer review history of their article (what does this mean?). If published, this will include your full peer review and any attached files.

Reviewer #1: No

---

## [Editor Report · Acceptance letter]

9 Mar 2020

PONE-D-19-20368R1 

Knowledge and remaining gaps on the role of animal and human movements in the poultry production and trade networks in the global spread of avian influenza viruses – a scoping review 

Dear Dr. Hautefeuille:

I am pleased to inform you that your manuscript has been deemed suitable for publication in PLOS ONE. Congratulations! Your manuscript is now with our production department. 

With kind regards,

on behalf of

Dr. Charles J. Russell 

Academic Editor

PLOS ONE